# Phosphorylation of LKB1 by PDK1 Inhibits Cell Proliferation and Organ Growth by Decreased Activation of AMPK

**DOI:** 10.3390/cells12050812

**Published:** 2023-03-06

**Authors:** Sarah Borkowsky, Maximilian Gass, Azadeh Alavizargar, Johannes Hanewinkel, Ina Hallstein, Pavel Nedvetsky, Andreas Heuer, Michael P. Krahn

**Affiliations:** 1Medical Cell Biology, Medical Clinic D, University Hospital of Münster, Albert-Schweitzer Campus 1-A14, 48149 Münster, Germany; 2Institute of Physical Chemistry, University of Münster, Corrensstr. 28/30, 48149 Münster, Germany

**Keywords:** LKB1, PDK1, AMPK, mTOR, cell proliferation

## Abstract

The master kinase LKB1 is a key regulator of se veral cellular processes, including cell proliferation, cell polarity and cellular metabolism. It phosphorylates and activates several downstream kinases, including AMP-dependent kinase, AMPK. Activation of AMPK by low energy supply and phosphorylation of LKB1 results in an inhibition of mTOR, thus decreasing energy-consuming processes, in particular translation and, thus, cell growth. LKB1 itself is a constitutively active kinase, which is regulated by posttranslational modifications and direct binding to phospholipids of the plasma membrane. Here, we report that LKB1 binds to Phosphoinositide-dependent kinase (PDK1) by a conserved binding motif. Furthermore, a PDK1-consensus motif is located within the kinase domain of LKB1 and LKB1 gets phosphorylated by PDK1 in vitro. In *Drosophila*, knockin of phosphorylation-deficient LKB1 results in normal survival of the flies, but an increased activation of LKB1, whereas a phospho-mimetic LKB1 variant displays decreased AMPK activation. As a functional consequence, cell growth as well as organism size is decreased in phosphorylation-deficient LKB1. Molecular dynamics simulations of PDK1-mediated LKB1 phosphorylation revealed changes in the ATP binding pocket, suggesting a conformational change upon phosphorylation, which in turn can alter LKB1’s kinase activity. Thus, phosphorylation of LKB1 by PDK1 results in an inhibition of LKB1, decreased activation of AMPK and enhanced cell growth.

## 1. Introduction

The serine-threonine kinase LKB1 was originally identified in Caenorhabditis elegans as the “Partitioning defective protein” 4 (Par4) and described to be essential for asymmetric division in *C. elegans* zygotes [1,2]. Similar, *Drosophila* LKB1 (DmLKB1) determines anterior-posterior polarity of the oocyte and apical-basal polarity in epithelial cells of the follicular epithelium and in the compound eye [3,4]. In *Drosophila* neural stem cells (neuroblasts, NBs), DmLKB1 regulates spindle formation and asymmetric cell division [5] and has been identified as an upstream regulator of the Hippo pathway effector Yorkie [6]. LKB1 is a master kinase, activating several downstream kinases, in particular of the AMPK (AMP-dependent kinase)-family, e.g., AMPK, MARKs, SAD-Kinases and NUAKs [7]. Furthermore, LKB1 has been shown to activate the tumor suppressors PTEN and p53 [8,9]. Thereby, LKB1 regulates various cellular processes, including cell polarity, cell migration, cell cycle control, apoptosis and energy metabolism (reviewed by [10]).

Mutations in STK11, the gene encoding human LKB1 (hLKB1) are the cause of the Peutz–Jeghers Syndrome (PJS), a rare autosomal dominant disease, which manifests in intestinal benign polyps and mucocutaneous mispigmentations/lentigines [11,12]. Patients suffering from PJS exhibit a strongly increased risk to develop intestinal and extraintestinal cancer. Moreover, LKB1 has been demonstrated to be mutated or downregulated in various tumor types, in particular in non-small cell lung cancer, prostate cancer and cervix carcinoma (reviewed by [10,13]). In mice models, inactivation of LKB1 together with mutations in PTEN results in a strongly enhanced tumorigenic potential (reviewed by [14]). Although LKB1 has been supposed to be a constitutively active kinase, its export from the nucleus and its activity are enhanced upon binding to the pseudokinase STRADα and to the adaptor protein Mo25 [15,16,17]. Apart from formation of this trimeric complex, several posttranslational modifications have been found to regulate the activity of the LKB1 complex [18]. We recently found that LKB1 binds directly to phosphatidic acid (PA) via a C-terminal polybasic motif. This motif is essential for stable membrane recruitment of LKB1 in cultured cells and in vivo. Furthermore, binding to PA is essential for efficient kinase activity and for the tumor suppressor function of LKB1 [19].

Activation of protein kinases by phospholipids of the (plasma) membrane has been described for several enzymes, including mTOR, SGK3, PKCs and Raf1 [20,21,22,23,24,25]. Phosphoinositide-dependent kinase 1 (PDK1) contains a pleckstrin homology (PH) domain, which binds to Phosphatidylinositol(3,4,5)-tris-phosphate (PI(3,4,5)P3), thereby releasing autoinhibition of the enzyme, resulting in an activation of PDK1 [26]. PDK1 activates several downstream serine/threonine kinases of the AGC family [27], in particular Akt/PKB, which is further activated by PIP3 [28], controlling mTOR signaling and other important cellular signaling pathways [29,30].

In this study, we now describe a new regulatory mechanism of LKB1 in *Drosophila*: The binding to and direct phosphorylation of a conserved motif within its kinase domain by PDK1 inhibits the activity of LKB1, resulting in decreased activation of AMPK. This was approved using molecular dynamics (MD) simulations, showing that the size of the binding pocket shrinks in the case of the phosphorylated protein. Cells expressing phospho-deficient LKB1 consequently display enhanced AMPK activation and decreased mTOR activity, resulting in reduced cell size.

## 2. Materials and Methods

### 2.1. Drosophila Stocks and Genetics

Fly stocks were cultured on standard cornmeal agar food and maintained at 25 °C. Knockin of LKB1 variants was established using CRISPR/Cas9 technique. In short, a plasmid (pU6-Bbs-chiRNA) encoding the guide-RNAi (GTTATCATGAAGTGCAATCA) targeting Cas9 to the third intron of LKB1 was injected into vasa::Cas9 transgenic flies (#51323 obtained from Bloomington stock center) together with a donor plasmid containing ca. 1 kbp 5′ and 3′ homology arms and an eye-driven (3xP3 promoter) dsRed (pHD-dsRed) [31]. Point mutations (T353A and T353D) were introduced by site-directed mutagenesis. MARCM (mosaic analysis with a repressible cell marker) clones were produced by crossing LKB1 FRT82B flies with hsFlp, tub::GAL4, UAS::nGFP;;FRT82B, tubP::GAL80 (obtained from Bloomington Stock Center, Bloomington, IN, USA). GFP-marked LKB1-variant-mutant clones in imaginal discs were induced by heat shock in first instar larvae.

### 2.2. Immunohistochemistry

Imaginal discs of third instar larvae were dissected in PBS and fixed for 20 min in 4% PFA/PBS. Subsequently, discs were washed three times in PBS + 0.2% Triton X-100 and blocked with 1% BSA for 1 h, incubated over night with primary antibodies in PBS + 0.2% Triton X-100 + 1% BSA, washed three times and incubated for 2 h with secondary antibodies. After three washing steps and DAPI-staining, discs were mounted with Mowiol. Embryos were fixed and stained as described before [32]. Primary antibodies used were as follows: anti Baz (1:500) [32], goat anti GFP (1:500, #600-101-215, Rockland, Pottstown, PA, USA), mouse anti Disc Large (1:100, 4F3, Developmental Studies Hybridoma Bank (DSHB), Iowa City, IA, USA) and mouse anti Histone-3 phospho-S10 (1:500, Cell Signaling #9706, Danvers, MA, USA). Secondary antibodies conjugated with Alexa 488, Alexa 568 and Alexa 647 (Life Technologies, Carlsbad, CA, USA) were used at 1:400. Images were taken on a Leica SP8 confocal microscope (Leica Microsystems, Wetzlar, Germany) using lightning program and processed using ImageJ version 1.53t.

### 2.3. Coimmunoprecipitation and Western Blot

For coimmunoprecipitation, Schneider S2R+ cells, which were isolated from late-stage *Drosophila* embryos [33] and commonly used for protein expression were cotransfected with GFP-LKB1 variants and PDK1-HA. Three days after transfection, cells were lysed and GFP-LKB1 was immunoprecipitated using GFP-binder (Chromotek, Planegg, Germany). Embryonic lysates were made of *Drosophila* embryos collected from an overnight plate with Laemmli buffer. SDS PAGE and Western blotting was performed according to standard procedures. The following primary antibodies were used: mouse anti ß Actin (1:1000, Santa Cruz #47778, Dallas, TX, USA), rabbit anti phospho-AMPK T172 (1:500, Cell Signaling #2535), mouse anti AMPK (1:500, Santa Cruz #sc-74461), mouse anti HA (1:500, Santa Cruz #7392), mouse anti GFP (1:500, Santa Cruz #9996), guinea pig anti LKB1 (1:500), mouse anti Myc (1:100, 9E10, DSHB) and mouse anti pT389 S6K (1:500, Cell Signaling #9206).

### 2.4. In Vitro Kinase Assay

MBP-LKB1 and MBP-LKB1 E253A proteins were expressed in *E. coli* BL21 cells and purified using Amylose resin. In total, 2 µg of recombinant protein was incubated together with 1 µg recombinant PDK1 (ProQuinase #0367-0000-1, Malvern, PA, USA) and 0.3 µCi[γ-32ATP] in kinase buffer (10 mM HEPES pH 7.5, 100 mM NaCl, 10 mM MgCl_2_, 1 mM DTT) for 1 h at 30 °C. The reaction was terminated by addition of SDS sample buffer and samples were subjected to SDS-PAGE. Phosphorylation was detected by exposure to X-ray films.

### 2.5. Molecular Dynamics Simulations

Chain C of PDB 2WTK, corresponding to the LKB1 protein [34], was extracted using VMD [35]. The missing amino acids (75–77) were modeled using Modeller version 9.16 [36]. Then, CHARMM-GUI membrane builder [37] was used to prepare the phosphorylated and unphosphorylated proteins, which were then amidated and acetylated and finally solvated by water molecules with the addition of neutralizing ions.

The MD simulations were performed using version 2019.6 of GROMACS [38] and the CHARMM36 force field [39], as well as the TIP3P model for water molecules. Periodic boundary conditions were applied in all directions. The long-range electrostatic interactions were treated using particle mesh Ewald method [40], with a cutoff distance of 1.2 nm and a compressibility of 4.5 × 10^−5^.

For the van der Waals (vdW) interactions, cutoff schemes with a cutoff distance of 1.2 nm were utilized, smoothly truncated between 1.0 and 1.2 nm. Constant pressure was controlled by coupling the system to the Parrinello-Rahman barostat [41] with an isotropic pressure of 1 bar. The temperature was controlled at 310 K by coupling the system to the Nosé-Hoover thermostat [42]. The LINCS algorithm was employed to constrain the bonds [43]. The systems were first minimized in 10,000 steps and were subsequently equilibrated using initially the NVT (500 ps) and then the NPT (16 ns) protocol in multiple steps. During the course of equilibration, restraints (starting with 4000 kJ/mol^−1^.nm^−2^) were applied on the heavy atoms of the protein, which were then gradually decreased to zero. The production simulations for both phosphorylated and unphosphorylated proteins were performed for 3 μs using a time step of 2 fs. To judge the statistical relevance of our results, we performed two independent runs, denoted sample 1 and sample 2.

The simulations data were analyzed using in-house codes, incorporating the MDAnalysis package [44]. VMD was used to visualize the structures and trajectories as well as preparing snapshots [35].

### 2.6. Statistics

Statistical significance was determined by one way ANOVA with Bonferroni multiple comparison test: **** *p* < 0.0001, *** *p* < 0.001, ** *p* < 0.01, * *p* < 0.05, n.s. is not significant.

## 3. Results

### 3.1. LKB1 Contains a Canonical PDK1-Binding and -Consensus Motif

LKB1 is localized to the plasma membrane of epithelial cells and neural stem cells [4,19]. In contrast to many other kinases, LKB1 is not activated by T-loop phosphorylation but was supposed to be constitutively active. However, we have shown that direct binding of LKB1 to phosphatidic acid in the plasma membrane is essential for membrane recruitment and activation of LKB1 [19]. We now investigated whether plasma membrane-bound LKB1 gets modified by kinases, which localized to the plasma membrane, too. Checking the amino acid sequence of LKB1 for kinase consensus motifs, we identified a canonical PDK1-binding motif as well as a PDK1 consensus motif [45] in the kinase domain of LKB1 (Figure 1A). Notably, both motifs are highly conserved from fly to men, but not in *C. elegans*. Using transfected Schneider 2R+ (S2R+) cells [33], we verified that LKB1 co-immunoprecipitates with PDK1 and that mutation of the PDK1-binding motif (E253A) strongly decreases the interaction of the two proteins (Figure 1B). Mutation of the phosphorylated Threonine 353 to Alanine increases binding of PDK1 (Figure 1C), suggesting that LKB1-PDK1 binding is reinforced if phosphorylation cannot occur. Finally, in vitro kinase assay with recombinant MBP (maltose binding protein)-LKB1 demonstrates that PDK1 efficiently phosphorylates LKB1, but not LKB1 T353A (Figure 1D).

### 3.2. Phosphorylation of LKB1 by PDK1 Does Not Affect Protein Localization In Vivo

In order to test whether phosphorylation of LKB1 by PDK1 alters the localization of LKB1, we established phosphorylation-deficient (T353A) and phospho-mimetic (T353D) rescue constructs of GFP-LKB1 expressed from its endogenous promoter [19]. As depicted in Figure 2A–C, both mutant variants localize to the lateral plasma membrane, colocalizing with Discs large (Dlg) and indistinguishable from wild type GFP-LKB1. Bazooka (Baz) was used as a marker for the apical cell-cell contacts [32].

### 3.3. T353 Phosphorylation Is Not Essential for Survival of the Fly, but Regulates Organism Size

Knockout of LKB1 is pupal lethal [46] and deletion of LKB1 results in strong polarity defects in various cell types [3,4,5,6,47]. Therefore, we investigated whether PDK1-mediated phosphorylation of LKB1 is lethal and produces any (polarity) phenotypes. For that, we established wild type, phosphorylation-deficient and phosphomimetic knockins using CRISPR-Cas9 gene editing. Notably, we found no increased lethality and normal (or even better) adult flies hatching rates in both mutant variants (Figure 2D). Moreover, apical-basal polarity in epithelial cells and neuroblasts as well as anterior-posterior polarity in oocytes is not affected in mutant knockins (data not shown). However, quantification of body size of LKB1-alleles revealed a significant reduction in phospho-deficient knockin flies, suggesting changes in the function of LKB1 which affects cell proliferation, and thus, organ/organism growth (Figure 2E).

### 3.4. Modeling of T353 Phosphorylation Reveals a Narrowed ATP-Binding Pocket

As no crystal structure of *Drosophila* LKB1 has been resolved so far, we used human LKB1 to model the impact of LKB1 phosphorylation by PDK1. The kinase domain of *Drosophila* and human LKB1 is well conserved (67.9% identical acids, 82.5% similarity). Although located in the kinase domain, analysis of the published crystal structure of human LKB1 [34] suggests that the PDK1-phosphorylation site is not engaged in the catalytic center but that it is rather exposed at the surface of the LKB1/STRADα/Mo25 complex. In order to identify conformational changes upon T353 phosphorylation (T230 in human LKB1), we performed two independent atomistic simulations for the non-phosphorylated (LKB1) and phosphorylated (pLKB1) form of LKB1, with a total simulation time of 3 μs for each system. The root-mean-square deviations (RMSDs) of the backbone atoms of the protein shows that the structure of the protein remains stable throughout the simulation in both simulations (Appendix A). The RMSD for LKB1 and pLKB1 in the first sample are nearly constant between 0.7 and 2.3 μs. In the second sample, the RMSD is more stable for both LKB1 and pLKB1 proteins compared to the first sample. However, the LKB1 undergoes more structural changes between 1 and 1.7 μs (Appendix A). This is mainly due to the structural changes in the activation loop (A-loop) region, which is comparatively much more distorted compared to the first sample. Therefore, the main changes in the protein structure in the second sample is due to the changes in the A-loop region.

The root-mean-square-fluctuations (RMSFs) or atomic positional fluctuations describe how flexible the individual residues are. The average RMSFs of protein Cα atoms over the two samples shows that the residues 118–125 and 203–211 (in the A-loop region) are more flexible in the LKB1 compared to pLKB1 (Figure 3A and Appendix A), revealing distinct differences between the two proteins flexibility, in particular in the A-loop part. We further calculated the average structure from the trajectory between 1 and 3 µs for both LKB1 and pLKB1 systems by overlaying the structures in each frame on the crystal structure and then averaging over all frames. The overlay of the average structures for both systems is shown in Figure 3B. Quantification of the distance between the Cα atoms after optimum mapping reveals small but significant effects in amino acids 57–63 and 118–125 on the N-lobe, 203–211 on the A-loop and 223–230 and 253–273 on the C-lobe (Figure 3C). Notably, most of these regions are not close to the T353 phosphorylation.

These results suggest that there is a small but significant effect of phosphorylation propagating throughout the protein.

As part of the structural changes in the N-lobe region, the distance between S60 and A195 (located on the A-loop) become smaller in pLKB1, which occurs between 0.9 and 1.9 μs of the simulation (Figure 3D). Accordingly, the volume of the ATP-binding pocket, which was calculated using CASTp [48]⁠, is temporarily decreased in pLKB1 (Figure 3E). In the second simulation, the distance between these two residues remains nearly similar (Appendix A). Instead, the distance between K78 and E98, which point towards the binding pocket, is smaller for pLKB1 (Figure 3F), again narrowing the ATP- binding pocket (Figure 3G). This shows that the ATP-binding pocket can shrink in different ways upon T353 phosphorylation, suggesting a decreased kinase activity of LKB1.

### 3.5. LKB1 T353 Phosphorylation Regulates Kinase Activity and Cell Growth In Vivo

Next, we investigated whether the prediction drawn from the modeling simulations is recapitulated in vivo and accounts for the decreased body size of T353A knockin flies. Indeed, activation of AMPK (by phosphorylation of the T-loop by LKB1) is increased in lysates from T353A knockin embryos, whereas it is decreased in case of T353D (Figure 4A). Consequently, phospho-S6K, a downstream target of mTOR is decreased in T353A, indicating that an enhanced activation of phospho-deficient LKB1 results in increased AMPK activity, which inhibits mTOR signaling. Of note, in in vitro kinase assays, LKB1 T353A and LKB1 E253A did not display an altered kinase activity towards AMPK (Figure 4B), suggesting that the phosphorylation of LKB1 T353 by PDK1 is indeed essential for regulating the kinase activity. To further validate this finding in vivo, we generated clone mutants for T353wt, T353A and T353D, respectively, in an otherwise wild type background in wing imaginal discs using the MARCM (mosaic analysis with a repressible cell marker [49]) technique. Indeed, the cell size of mutant (GFP-marked) cells was reduced in case of T353A (Figure 4C–F). In addition, the percentage of proliferating cells (quantified by Histone 3 phospho-S10) was lower in T353A clones (0.9%), compared to wild type (1.2%) and T353D clones (1.4%). These results support our hypothesis derived from modeling simulations that phosphorylation of LKB1 by PDK1 at T353 inhibits the kinase activity of LKB1, thus resulting in decreased activation of AMPK and its downstream target mTOR, which leads to increased cell size and proliferation. We finally tested whether impaired LKB1 T353 phosphorylation results in changes in binding of the LKB1 cofactors Stlk and Mo25, which might explain differences in the activation of LKB1. However, no differences in Stlk/Mo25 binding were detectable in co-immunoprecipitation assays (Figure 4G), which is in line with the prediction from LKB1’s crystal structure that the binding interface to its cofactors does not involve T353.

## 4. Discussion

In this study, we describe the phosphorylation of the tumor suppressor kinase LKB1 by PDK1 as a new regulatory mechanism to control LKB1’s activity. LKB1 exhibits a conserved PDK1-binding motif, which is essential for binding to PDK1 as well as a conserved PDK1-phosphorylation motif within its kinase domain. Our modeling results suggested a decrease in LKB1 kinase activity due to narrowing of the activation loop in the catalytic center upon phosphorylation by PDK1. Indeed, we confirmed that a phosphorylation-deficient variant of LKB1 exhibits an increased activation of AMPK, the major substrate of LKB1, which results in increased mTOR activation and decreased cell proliferation. Notably, *C. elegans* Par4, the homologue of human and *Drosophila* LKB1 [1,2] exhibits a well conserved PDK1-binding and -phosphorylation motif but lacks the phosphorylated residue itself (T353 in *Drosophila*, F365 in *C. elegans*). This indicates a conserved regulatory mechanism, which was partly lost during evolution of nematodes. However, PDK1 has also been described to exhibit kinase-independent functions [50,51,52]; thus, the PDK1/LKB1 interaction might also be of importance in *C. elegans*.

In *Drosophila* in vivo, flies can obviously scope with impaired T353 phosphorylation, as overall development and hatching rates are not impaired. However, adult flies exhibit a reduced body size due to decreased cell proliferation upon overactivation of the AMPK/mTOR axis, which controls cell growth and proliferation [53]. Thus, in their physiological environment, the regulation of LKB1 activity by PDK1 might turned out to be a selection advantage during evolution.

Of note, mimicking a constitutive phosphorylation of LKB1 (T353D) displays identical phenotypes as wild type LKB1. This suggests that under cellular conditions of PDK1 activation, e.g., by PI3K activation upon growth factor stimulation, LKB1 is mainly phosphorylated by PDK1, thus providing a negative feedback mechanism, inhibiting aberrant mTOR activation by PI3K/PDK1 via the LKB1-AMPK axis.

Up to now, several upstream phosphorylation sites of LKB1 have been described to fine-tune the activity of LKB1: conserved residues at the very C-terminus (S562 in *Drosophila*, S428 in human LKB1) can be phosphorylated by PKA as well as aPKC, thereby promoting nuclear export, and thus, activation of LKB1 [4,54]. LKB1 Tyrosine phosphorylation by Fyn results in a similar activation [55], whereas phosphorylation of LKB1 by Aurora-A blocks binding and activation of AMPK, thus inhibiting the LKB1/AMPK axis [56].

The phosphorylation of LKB1 by PDK1 described in this study adds another upstream regulatory mechanism of LKB1, which is likely to be important for the finetuning of LKB1’s activity during development. Furthermore, this signaling pathway might serve as a backup to compensate increased PI(3,4,5)P3 levels in the plasma membrane, e.g., in PI3K gain of function or PTEN loss of function mutations, which occur in various types of cancer: increased PI(3,4,5)P3 enhances the activation of PDK1/Akt, which in turn results in the activation of mTOR [57], which is counterbalanced by simultaneous activation of LKB1/AMPK and subsequent inhibition of mTOR by AMPK [58,59]. In tumors, aberrant PDK1 activation by enhanced production of PI(3,4,5)P3 due to mutations in PI3K or PTEN frequently coincidences with downregulation of mutation of LKB1 [13,14], resulting in a decreased activation of AMPK, further enhancing Akt/mTOR activation, and thus, cancer progression.

PDK1 has been well characterized to phosphorylate and, thereby, activate kinases of the AGC family [60,61,62,63,64]. To our knowledge, LKB1 is only the third non-AGC kinase substrate, apart from the kinases p21-activated kinase (PAK1 [65]) and polo-like kinase [66] and Integrin β3 [67,68] to be phosphorylated by PDK1. Both kinases, LKB1 and PDK1, are recruited to the plasma membrane by direct binding to phospholipids: LKB1 binds to phosphatidic acid via its C-terminal polybasic motif [19], while PDK1 contains a PH domain, which preferentially recognizes PI(3,4,5)P3 [69]. Thus, the plasma membrane might serve as a platform for the LKB1-PDK1 interaction and the regulation of LKB1 by PDK1 phosphorylation in a similar mechanism as described for PDK1/Akt [70]—although in that case, both enzymes bind to the same phospholipid (PI(3,4,5)P3).

## Figures and Tables

**Figure 1 cells-12-00812-f001:**
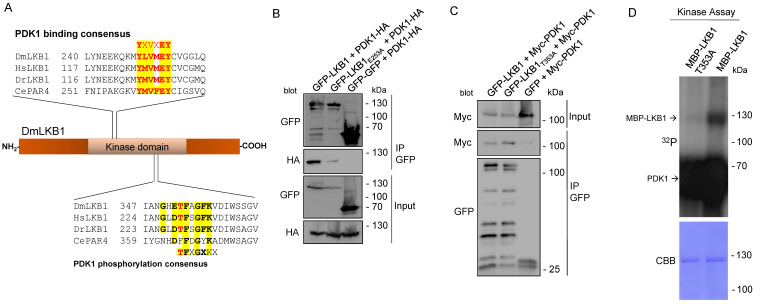
LKB1 displays conserved PDK1-binding and -phosphorylation motifs and is phosphorylated by PDK1 in vitro. (**A**) Scheme of *Drosophila* LKB1 and sequence alignment of PDK1-binding and consensus-motif. DmLKB1—*Drosophila melanogaster* LKB1, HsLKB1—*Homo sapiens* LKB1, DrLKb1—*Danio rerio* LKB1 (zebrafish), CeLKB1—*C. elegans* LKB1. (**B**,**C**) S2R+ cells were co-transfected with HA-PDK1 and either wild type GFP-LKB1 and GFP-LKB1 E253A (**B**) or wild type GFP-LKB1 and GFP-LKB1 T253A (**C**) and GFP alone. GFP-proteins were immunoprecipitated and GFP(-LKB1) and HA-PDK1 (**B**) or Myc-PDK1 (**C**) were detected by immunoblotting. (**D**) Recombinant wild type or mutant MBP-LKB1 produced in *E. coli* was used in a 32P radioactive kinase assay with recombinant PDK1. CCB is colloidal coomassie blue, which was used to visualize the LKB1 input proteins.

**Figure 2 cells-12-00812-f002:**
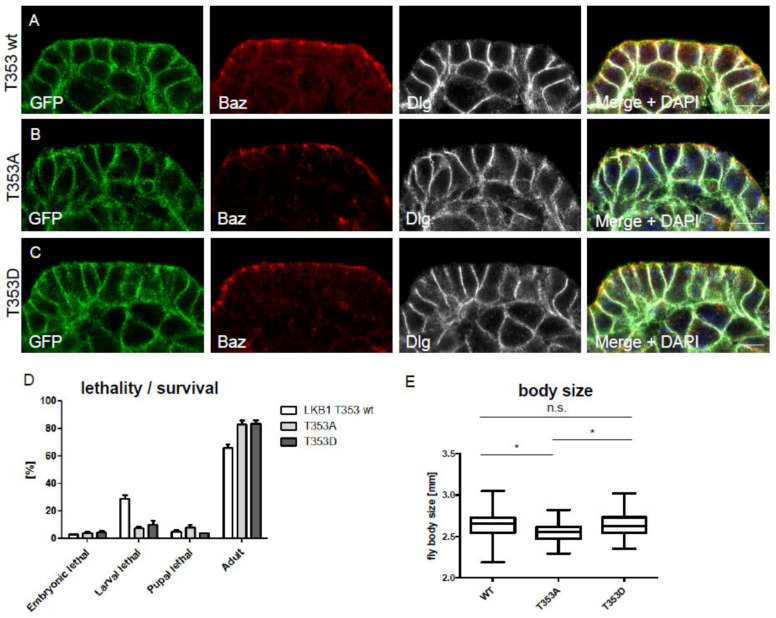
GFP-LKB1 variants localize normally and do not impair fly survival but affect body size (**A**–**C**) Immunostainings of embryonic epidermis epithelial cells expressing GFP-LKB1 variants from its endogenous promoter. Discs large (Dlg) was used as marker for the lateral membrane and Bazooka (Baz) marks the apical junctions. (**D**) Flies with CRISPR/Cas9-mediated knockin of wild type LKB1, LKB1 T353A or LKB1 T353D display comparable survival rates (*n* = 100, *N* = 3), but different body sizes (**E**), *n* ≥ 35. Scales bars are 5 µm in (**A**–**C**). Error bars are standard error of the means. Significance was determined by one way ANOVA with Bonferroni multiple comparison test: * *p* < 0.05, n.s. not significant.

**Figure 3 cells-12-00812-f003:**
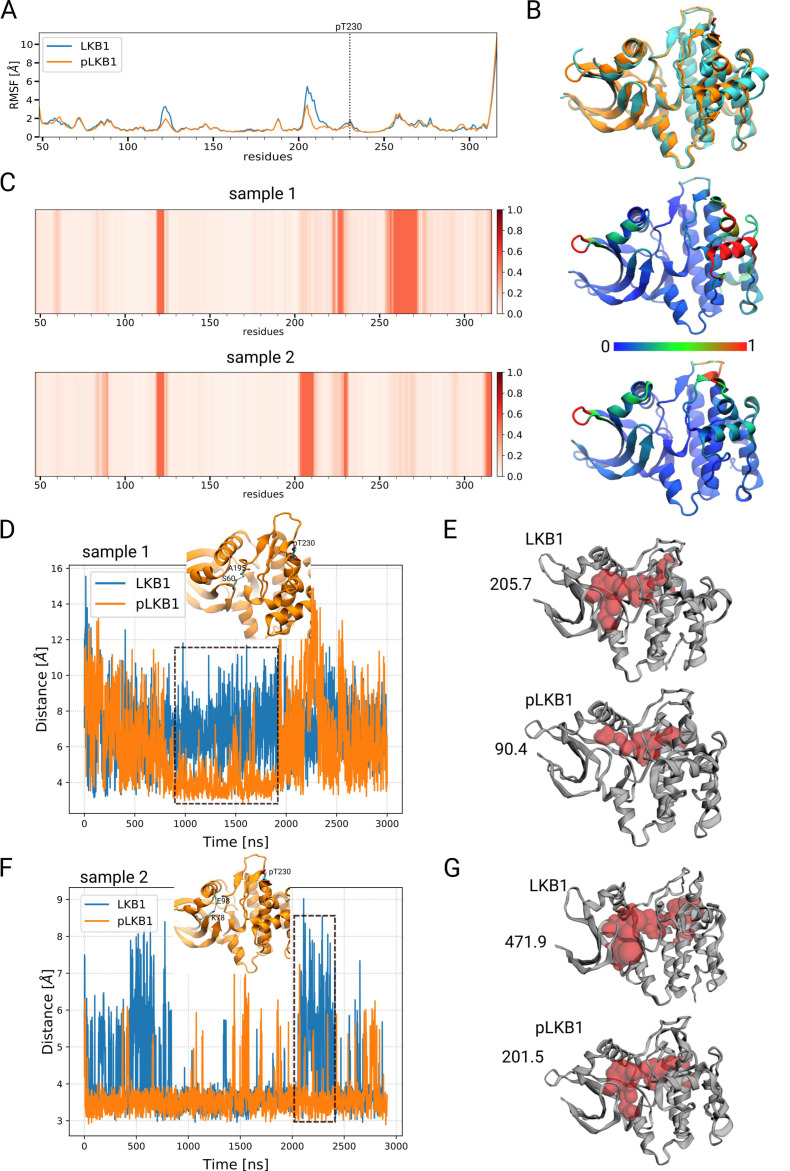
Simulation of LKB1 phosphorylation reveals changes in the ATP-binding pocket. (**A**) The RMSF of the protein Cα atoms averaged over the two samples is shown for human LKB1 (Blue) and human phospho-LKB1 (pLKB1, red). (**B**) Overlay of the two average structures for human LKB1 (blue) and human pLKB1 (orange). The average structures were obtained from 1 to 3 μs of the simulations. (**C**) The difference between human LKB1 and human pLKB1 average structures are shown for the two samples. For each Cα atom a shift distance is determined. The atoms with the 50% largest distances are described by a shifted value of 1. The remaining distances are ordered and linearly mapped on shift values between 0 and 1. The shift values are translated into the respective color codes. This non-linear procedure avoids that the parts, which are dramatically changed, blur the other parts. On the right-hand side, the cartoon representation of the phosphorylated protein is shown, colored based on the right-hand side figure, showing the degree of change in different parts of the protein between the two systems. (**D**) The distance between S60 and A194 residues over the simulation time. The protein is drawn and these two residues, along with the pT230 (corresponding to T353 in *Drosophila* LKB1) residue, are shown as stick representations. (**E**) The free volume available inside the ATP-binding pocket, obtained from 0.9 to 1.9 μs (represented with a dashed rectangle) of the first sample simulations, was calculated using CASTp and is shown in red spheres. (**F**) The distance between K78 and E98 residues over the simulation time along with the pT230 residue are shown in stick representations. (**G**) The free volume available inside the protein for the average structures, obtained from 2.0 to 2.4 μs (represented with a dashed rectangle) of the second sample simulations.

**Figure 4 cells-12-00812-f004:**
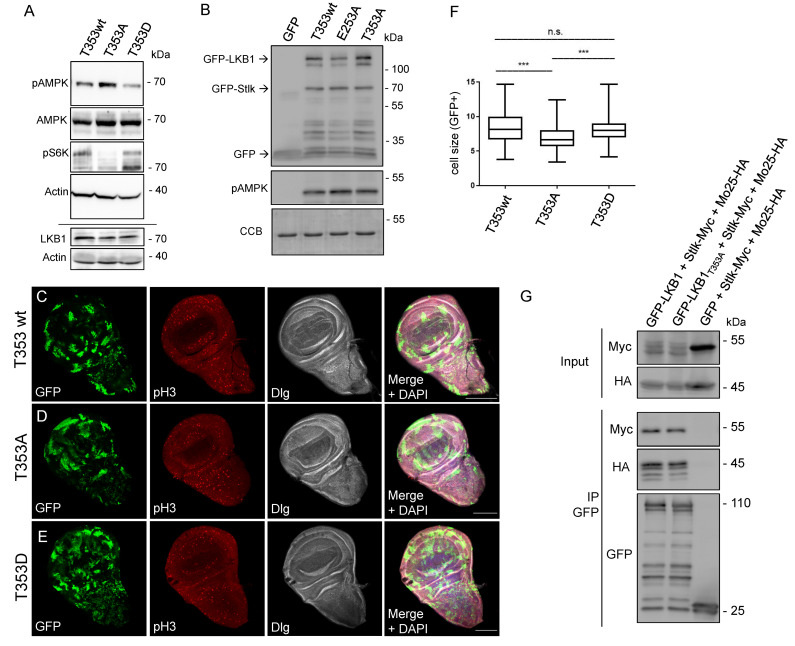
LKB1 T353 phosphorylation regulates AMPK activation, cell size and proliferation in vivo. (**A**) Western blots of embryonic lysates of LKB1 wt, T353A and T353D knockin flies against the indicated proteins. (**B**) In vitro kinase assay with recombinant GST-AMPK (aa 108–280) and the indicated LKB1 variants plus its cofactor Stlk. CCB is colloidal coomassie blue, which was used to visualize the GST-AMPK input proteins. (**C**–**F**) GFP-marked MARCM (mosaic analysis with a repressible cell marker) clones of LKB1 knockin-variants in otherwise wild type tissue in wing imaginal discs stained with a proliferation marker (Histone 3 phospho-S10, pH3) and Dlg to label cell boundaries. The size of GFP-marked cells was quantified (*n* > 200). Scale bars are 20 µm. Error bars are standard error of the means. Significance was determined by one way ANOVA with Bonferroni multiple comparison test: *** *p* < 0.001, n.s. not significant. (**G**) S2R+ cells were co-transfected with Mo25-HA, Stlk-Myc and either wild type GFP-LKB1 or GFP-LKB1 T353A. GFP alone was used as control. GFP proteins were immunoprecipitated and GFP(-LKB1), Mo25-HA and Stlk-Myc were detected by immunoblotting.

## Data Availability

All data are available in main and Appendix A.

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
