# Peer review of "Phosphorylation of LKB1 by PDK1 Inhibits Cell Proliferation and Organ Growth by Decreased Activation of AMPK"

_cells, 2023, doi:10.3390/cells12050812_

Round 1
Reviewer 1 Report
Comments to Authors
This study by Borkowsky et al demonstrates a novel regulatory mechanism of LKB1 by PDK1-mediated phosphorylation in vitro and in genetically engineered Drosophila. First the authors show that the binding of the kinase domain in LKB1 with PDK1 and phosphorylation at Thr353 in LKB1 by PDK1. The authors further simulate the effect of Thr353 phosphorylation on the structure of the ATP-binding pocket of LKB1, suggesting a decrease kinase activity. They finally show that the body size is significantly reduced in phosphorylation-deficient (T353A) LKB1 knockin flies consistent with the reduction of cell size of the mutant cells. The results also demonstrate the significant induction of AMPK phosphorylation in embryonic lysate of T353A knockin flies that participates in inhibition of mTOR signaling.
It is a nice and clean study. The experiments are well performed and the results are well analyzed. The discussion nicely goes to the important points brought up by the work. Overall, this study provides essential new information on the novel molecular mechanisms regulating LKB1/AMPK pathway by PDK1. However, I have the following comments and suggestions that should be addressed.
1. The authors demonstrate LKB1 is capable of interacting with PDK1 by using IP assay from overexpressing S2R+ cells (Fig. 1B). Since the interaction of both kinases is apparently stable, the authors should demonstrate the interaction of endogenous kinases.
2. The authors suggest a reduced LKB1 activity by PDK1-catalyzed phosphorylation at T353 based on molecular dynamics simulations. It would have been nice to show this by using in vitro kinase assay, because the authors could successfully obtain phosphorylated MBP-LKB1 (at T353) by PDK1 (Fig. 1 C) and T353A mutant, which could be used for evaluating the effect of the phosphorylation at Thr353 on the LKB1 activity in vitro.
3. As shown in Fig. 2 and Fig. 4, similarity of phenotypes between wild type and phospho-mimetic T353D knockin flies as compared with phospho-deficient (T353A) knockin flies including body size, cell size and the phosphorylation levels of AMPK and S6K should be explained. Does it mean LKB1 is constitutively phosphorylated at Thr353 in Drosophila? If so, the authors should demonstrate Thr353 phosphorylation of LKB1 in wild type knockin flies.
4. Western blot for LKB1 should be included in Fig. 4A.
Minor comments:
5. PDK1 binding consensus in Fig. 1A is YXVXEY not FXXFEY according to the sequence comparison. The authors have to add its related reference(s) and explain abbreviations for species in the figure legend.
6. PDK1 phosphorylation consensus is TFXGXK not TFCGTXDY according to the sequence comparison. The authors also have to add its related reference(s).
7. Typos in Fig. 1 B and C. 
8. References should be numbered (line 147, 151, 171 ; Dogliotti et al., 2017, line 175; Kullmann and Krahn, 2018c).
9. In page 4, line 158, E252A should be E253A.
10. The authors have to describe statistical analysis in “Materials and Methods” as well as in figure legends.
11. Please make sure to use the same font and font size throughout the manuscript.
Reviewer 2 Report
In this manuscript, the authors present data supporting a novel role for PDK1 in down-regulation of LKB1, with consequent decreased AMPK activity. The finding could be interesting, but some points need to be better clarified.
Major points:
1. Throughout the work, there is confusion between human and Drosophila LKB1; all the experimental studies have been performed on the Drosophila protein, but the effect of phosphorylation has been modeled on the human one.
Which is the percentage of identity between human and Drosophila LKB1? (The homology is shown only for the short regions encompassing the PDK1 binding and phosphorylation sites).
The authors should at least comment on the extent to which they expect their results can be exported to human cells.
2. LKB1 displays consensus for both PDK1 binding and phosphorylation. It should be assessed if the two events are correlated or independent, and which one is more relevant for the regulation of LKB1 function (also considering the C. elegans sequence, lacking the phosphorylation site, and the observation that PDK1 has also kinase-independent functions, as the Authors mention). The employment of the LKB1 mutants could help in this analysis: is the E253A mutant phosphorylated by PDK1 or not? is the mutant defective of phosphorylation (T353A) still able to bind to PDK1? And does it form regular LKB1/STRADa/Mo25 complex?
3. Does the genetic or pharmacological down-regulation of PDK1 recapitulate the effects of LKB1 T353A? This would strongly corroborate the findings of this work
Minor points:
1. S2R+ cells should be better defined. Also, what does “embryonic lysates” mean?
2. In Fig 1A, please define the organisms of the different sequences
2. It could be interesting to discuss the possible impact on AMPK signaling of conditions where PDK1 is hyperactivated (via canonical and non-canonical mechanisms)
Round 2
Reviewer 1 Report
I have no further concerns of the manuscript. It is now a very nice study ready for publication.
Reviewer 2 Report
In this new version of the manuscript, the Authors have addressed most of my points, providing reasonable justifications for the experiments they did not perform. The level of the manuscript is significantly improved.